# New Isolation of Ponticelli III Virus (*Bunyavirales*: *Phenuiviridae*) in Emilia-Romagna Region, Italy

**DOI:** 10.3390/v15020422

**Published:** 2023-02-02

**Authors:** Mohamed Daoudi, Giuseppe Romeo, Katia Marzani, Angelica Petrella, Paolo Bonilauri, Davide Lelli, Ali Boumezzough, Samia Boussaa, Michele Dottori, Mattia Calzolari

**Affiliations:** 1Microbial Biotechnologies, Agrosciences and Environment Laboratory (BioMAgE), Faculty of Sciences Semlalia, Cadi Ayyad University, Marrakesh 40000, Morocco; 2Istituto Zooprofilattico Sperimentale della Lombardia e dell’Emilia-Romagna “Bruno Ubertini”, 42124 Brescia, Italy; 3Higher Institute of Nursing and Technical Health Occupations (ISPITS), Ministry of Health, Rabat 10000, Morocco

**Keywords:** ponticelli III, phleboviruses, circulation, Emilia-Romagna, Italy

## Abstract

The number of newly described sandfly-borne phleboviruses has been steadily growing in recent years. Some phleboviruses are human pathogens, but their health relevance is largely uncharacterized. We aimed to investigate the circulation of these viruses in the Emilia-Romagna region where several have already been described. A total of 482 sandflies were collected in a site in Reggio Emilia in 2019 and 2020. Sandflies collected in 2020 were grouped in 21 pools with a maximum of 25 sandflies per pool, submitted to real time PCR, and isolated in Vero cell culture. Complete genome sequencing showed the isolation of a strain of a Ponticelli III virus. This virus, which belongs to the species *Adana phlebovirus*, differed in the M segment from the Ponticelli I and Ponticelli II viruses. Analysis performed on the genomic segments of the newly isolated virus compared with other phleboviruses highlighted a strong purifying selection in the L segments, and different substitution saturation, highest in the M segments. Future research should address the ecological processes driving the occurrence of these novel phleboviruses and their possible impact on public health.

## 1. Introduction

The family *Phenuiviridae* includes 20 genera, the members of three of which are largely transmitted by arthropods: *Phlebovirus*, *Uukuvirus* and *Ixovirus* [1,2]. The genus *Phlebovirus* comprises about 70 named viruses widely distributed in Europe, Africa, Central Asia and the Americas. As provided by the International Committee on Taxonomy of Viruses (ICTV), these viruses are currently classified into species according to the threshold of 95% identity in the amino acid sequence of the RNA-dependent RNA polymerase (RdRp) protein [3]. Some of these viruses are medically relevant, such as the Rift Valley virus (RVFV) transmitted by mosquitoes and the Toscana virus (TOSV). Other medically relevant viruses transmitted by sandflies include sandfly fever Naples and Sicilian phleboviruses in the Old World, and Alenquer, Candiru, Chagres, Cocle, Echarate, Maldonado, and Punta Toro phleboviruses in the New World [4].

Phleboviruses are single-stranded negative-sense RNA with a genomic organization consisting of three segments, designated S (small), M (medium), and L (large). The S segment bears two genes encoding, by an ambisense strategy, the nucleoprotein (Nc) and non-structural (Ns) protein; the M segment encodes for enveloping the glycoproteins; and the L segment for the RdRp [5,6]. Phleboviruses are morphologically similar, with a diameter of 80-120 nm, icosahedral symmetry, and an envelope formed by short, spike-like glycoproteins embedded in a lipid bilayer [7,8].

In the Old World, at least 250 million people have been exposed to *Phlebovirus* infections [9,10]. One of the main examples is TOSV, which is known as an emerging pathogen causing meningitis in the Mediterranean region, for which defined reservoirs have not yet been identified. It is unlikely that humans are the reservoir for TOSV because human viremia duration is too short [11].

Sandfly-borne phleboviruses’ distribution and circulation peaks are closely related to vector presence and activity [12]. Sandflies are broadly distributed in regions bordering the Mediterranean basin. They are often abundant in peri-urban, rural and natural environments, which are concentrated near human and domestic animal habitats in suitable areas [13,14].

Recently, several studies conducted in Mediterranean countries showed an increase in *Phlebovirus* circulation with newly discovered viruses, many of which were detected in sandflies [15,16]. One example is the Adana virus, found in the eponymous name of Adana province in Turkey. The DNA barcoding characterization of the positive pool showed that the pool from which the virus was isolated contained *Phlebotomus (Ph.) tobbi*, *Ph. perfiliewi*, and *Ph. papatasi* [17]. The genetic and phylogenetic analysis revealed that Adana virus is closely related to Salehabad virus and Arbia virus [17]. Conspecific viruses of Adana virus were successively isolated in Italy (Ponticelli I, Ponticelli II, Ponticelli III viruses) [18], in Croatia, and in Northern Macedonia (Bregalaka virus), however they showed different M segments with respect to Adana virus [19].

In our current study, we report a new isolation of a Ponticelli III virus (PONTV III) strain in Emilia-Romagna at about 80 km from the site of the first report of this virus in the region [18].

## 2. Materials and Methods

### 2.1. Sandf Fly Trapping and Identification

Samples were collected in June 2019 and September 2020 in Casalgrande locality (44°33′40.3″ N, 10°43′27.8″ E), Reggio Emilia province. The site was located in a hilly area of the Emilia-Romagna region at 257 m altitude (Table 1). Sandfly sampling was performed overnight using two types of traps: carbon-dioxide-baited traps and CDC light traps. In 2019, sandflies were morphologically identified with an optical microscope after clarification, while in 2020, the collected sandflies were stored in dry ice and used directly for virus isolation in cell culture with a maximum of 25 specimens per pool.

### 2.2. Virus Isolation on Cell Cultures

Female specimens pools were homogenized in minimal essential medium, supplemented with penicillin and streptomycin, using a sterilized glass potter and clarified by centrifugation at 3000× *g* for 15 min. Samples were inoculated in a confluent monolayer of VERO cells (African green monkey kidney cells) at passage 170 (cell culture bio bank of IZSLER, code BSCL86), incubated at 37 °C with 5% CO_2_ and observed daily for 7 days to observe the development of cytopathic effects (CPE) [18]. To exclude the presence of two or more viral strains in culture, selected isolates were duplicated by plaque purification in two rounds of serial dilutions in 96-well plates (maximum dilution of 10^−8^). One of the obtained duplicates was recultured on VERO cells and subjected to sequencing.

### 2.3. Virus Detection and Sequencing

Viral RNA was extracted from supernatant cultures that displayed CPE using TRIzol LS Reagent (Invitrogen, Carlsbad, CA, USA); cDNA synthesis was performed using random hexamers (Roche Diagnostics, Mannheim, Germany) and SuperScript™ II reverse transcriptase (Invitrogen, Carlsbad, CA, USA). Complementary DNA was submitted to PCR for the detection of phleboviruses (Pan-Phlebo PCR) using one reverse and two forward primers and targeting a 370-nucleotide region of the S segment [20]. The obtained amplicons were sequenced, and the sequences were utilized for virus identification by BLAST analysis against the GenBank (GB) database (https://blast.ncbi.nlm.nih.gov/Blast.cgi, accessed on 28 January 2023).

The selected viruses isolated from the cell culture supernatant were sequenced on a MiSeq Instrument (Illumina Inc., San Diego, CA, USA). The preparation DNA libraries for whole-genome sequencing and viral RNA amplification were performed following the Kampmann et al. (2011) protocol [21].

### 2.4. Phylogenetic Analysis

The complete genome of the segments was aligned with the homologous sequences of other phleboviruses available in GB. The nucleotide alignments were obtained using the translated amino acidic sequence with the Mega11 software [22] (muscle algorithm with default parameters) and refined manually. Obtained alignments were used to infer the maximum likelihood trees by IQ tree software, with the selection of the best fit model performed by the software and branch supports inferred with ultrafast bootstrap analysis [23]. Software RDP5 [24] was used to check possible recombination events between different viruses. High variability was recorded in the alignments, so to minimize the rate of false positive result we used the methods MAXCHI, GENECONV and CHIMAERA, which showed most optimal performance in this situation. Only potential recombination events detected by all three methods were considered as suitable [25]. Aligned nucleotide sequences of the different segments of the PONTV were analyzed with DAMBE software to obtain a saturation plot in which transitions and transversions were plotted against GTR evolutionary distance. This allowed us to evaluate the genetic saturation of the different segments [26]. The number of transitions and transversions versus divergence offered a visual display of substitution saturation. To evaluate per-site selection in the same nucleotide alignment, the FEL (fixed effects likelihood) method of data monkey (using a *p* value of 0.05) was used, FEL infers nonsynoymous (dN) and synonymous (dS) substitution rates using a maximum-likelihood (ML) approach for a given coding alignment [27]. Saturation plot and per-site selection of the two ambisense genes of the S segment were conducted by concatenating the two sequences in the same reading frame. The p-distances (the complement number of identity) of amino acid and nucleotide sequence) were computed with MEGA X software [28]. Graphs were visualized by the ggplot2 package in R.

## 3. Results

A total of 482 sandflies were collected (73 in 2019 and 409 in 2020) in Casalgrande locality, Emilia-Romagna region, Italy. Only some specimens collected in 2019 were identified using morphological characteristics; of these, 72 were *Ph. perfiliewi* and 1 was *Ph. perniciosus* (Table 1). The 135 collected sandflies in 2020 were submitted to viral isolation. Of the five 25-specimen pools and one 10-specimen pool of sandflies, only one pool showed a CPE on VERO cells (Table 1). CPE observed on VERO cells, from day 3 post-infection, was marked by diffuse degeneration of the cell monolayer, with progressive rounding and subsequent detachment of the cells which appeared numerous and fluctuating in the culture medium. Cell culture supernatants with CPE were subjected to Pan-Phlebo PCR, producing a sequence ascribable to a PONTV (Figure 1). This sample was subjected to complete genome sequencing.

According to the previously obtained sequences, and particularly the M sequence, the new isolate was classified as PONTV III, strain 270315. Obtained sequences were deposited in GB by the accession numbers OP293791 for the L segment, OP293792 for the M segment, and OP293793 for the S segment. These sequences showed high identity for the PONTV II 238134-4/2016 (MG911981) in the L segment (99.13%), PONTV III 238134-3/2016 (MG911977) in the M segment (99.46%), and PONTV III 195684-2/2016 (MG911986) for the S segment (98.33%).

Sequences of the three segments were used with homologous sequences of sandfly- borne phleboviruses from the Mediterranean and Middle East that were available in GB for phylogenetic analysis. Two alignments were obtained with L and M segments (Figure 1) and one with the Ns and Nc (Appendix A). The evolutionary model selected for inferring ML trees, according to the lowest Bayesian information criterion, was the transition model with empirical base frequencies and free rate heterogeneity (TIM2+F+R4) for the L alignment, the general time reversible model with empirical base frequencies and free rate heterogeneity (GTR+F+R4) for the M alignment, the transition model with empirical base frequencies and gamma rate heterogeneity (TIM2+F+G4) for the Nc alignment, and the TPM2u with empirical base frequencies and gamma rate heterogeneity (TPM2u+F+G4) for the Ns alignment. In the L, Ns and Nc trees, the Ponticelli viruses were in a well-supported clade, in the M tree the three viruses were in separate branches (Figure 1 and Appendix A).

None of the putative recombination events were detected by all three methods selected by RDP5 software in the alignments obtained by sequences of L and M segments.

The phylogenetic analysis obtained of L segments (Figure 1A) and the p-distance (Table 2) confirmed that our PONTV III virus, as for other PONTVs, belonged to *Adana phlebovirus* species, according to the 95% identity threshold. The phylogenetic tree obtained for M segments (Figure 1B) showed some topologic incongruences with respect to the L, Nc and Ns segment trees (Figure 1A and Appendix A), most likely due to reassortment events. Interestingly, among the viruses of the Salehabad complex employed in the analysis, the Bregalaka virus had an M segment strictly related to PONTV II (Table 2, Figure 1B). The overall mean nucleotide p-distance between the different Ponticelli viruses was 0.017 for the S segments, 0.020 for the L segments, and rose to 0.21 for the different M segments, while the p-value between the same type of M segments ranged between 0.015 (PONTV I, PONTV III) and 0.012 (PONTV II).

The FEL analysis highlighted a pervasive purifying selection in all sites: the 83% non-invariant sites under purifying selection in the L segment (1388 on 1673 non invariant sites tested), 63% in the S segment (293 on 464 non invariant sites tested), and 73% in the M segment (947 on 1295 non invariant sites tested). The FEL analysis identified only one of the sites under diversifying positive selection in segment M. Plotting nucleotide p-distance against aminoacid distance, we observed that the different segments seemed subject to a differential dynamic of evolution, with an increasing amino acidic and nucleotide diversity in M segments, while L segments showed less aminoacid diversity for the same amount of nucleotide diversity to the other segments (Figure 2). The analysis performed on the three PONTV demonstrated that purifying selection was more intense in the L segment, as demonstrated by the FEL analysis and the lowest aminoacid diversity at equal nucleotide diversity of the L segment, respectively, to M and S segments.

The saturation plot showed a higher relative number of transversions in the M segment, which exceeded that of transitions with the increase in GTR distance. This situation strongly suggested a substation saturation in the M segments (Figure 3).

## 4. Discussion

Detection of phleboviruses in hosts is made difficult by the overall diversity and genetic complexity of clinically relevant strains, their essentially undefined clinical associations, and the resulting lack of diagnostic tools [4]. A paradigmatic example is the Toscana virus (TOSV), the etiological agent of neuroinvasive diseases including meningitis and meningoencephalitis. This virus was first isolated in 1971 by Verani et al. (1982) [29] and only later recognized as an agent of meningitis in humans [30]. In addition to TOSV, Sicilian phlebovirus (SFSV), and Naples phlebovirus (SFNV) also have pathogenic potential for humans, which is generally mild fever [31]. Different species of vertebrates are presumed to be hosts for these viruses, but convincing evidence is lacking. Antibodies to TOSV and SFSV have been found in horses, cats, dogs, sheep, pigs, cattle, and goats [9,31]. TOSV has been isolated from brains of Kuhl bats (*Pipistrellus kuhlii*) in regions of Italy where several TOSV strains are circulating. Recently, positive sera of TOSV and SFSV measured using a seroneutralization test were detected in European bat colonies (free-tailed bats (*Tadarida teniotis*) and serovar bats (*Eptesicus serotinus*)) located in the Aragon and the Balearic Islands regions [10]. The possible infection of PONTV II was recorded by way of seroneuralization in Italy [30]. Moreover, *Bregalaka* virus (BREV) neutralizing antibodies were detected in 3.3% of human sera in North Macedonia, demonstrating that BREV frequently infects humans [10]. Interestingly, PONTV II and BREV were strictly related, having an identity of 96% in the amino acid sequence of the M segment. The clues of possible infectivity of these strictly related viruses for humans deserve more investigation.

In phleboviruses, the same L and S segments could be associated with different M segments. The presence of viruses differing in M segments was already recorded in two group of phleboviruses beyond the PONTV, Massilia, Arrábida and Granada viruses [32,33,34,35] in Europe and in the Candiru *phlebovirus* [36] in South America. This condition is recorded also in viruses of the *Orthobunyavirus* genus, in which reassortant viruses are common, but with the relevant difference that the donor virus of the M segment is often known [37]. The great genetic variability of phleboviruses is linked to the absence of the proofreading activity of the RdRp and counterbalanced by a strong and pervasive purifying selection able to preserve the amino acid sequence, which results more stable than the nucleotide sequence from which is translated, above all in the L segment.

Results obtained by the analysis of the PONTV indicated that the three segments were differentially saturated, with the M segment more variable and the L segment more preserved. More work is needed to understand if different saturations of the segments could be caused by a different level of purifying selection and consequent extinction of deleterious genotype, or if other genetic mechanisms are potentially creating this different evolutionary dynamic.

In the M segment, transversions occur more frequently than transitions, indicating a saturation of substitutions. The characterization of the mechanism at the base of this condition was hampered by the variability in M segment. While it is difficult to reconstruct recombination events that could have happened in the saturated condition of the M segment, we failed to detect a recombination signal between sequences of phleboviruses in this study. Even if it is not possible to completely exclude recombination events, this result argues in favor of the rarity of these events, as already highlighted in negative senses RNA viruses [38,39]. This suggests that a mechanism other than recombination can act to improve variability of the capsid genes. Reassortment is likely one of the more relevant mechanisms in producing the diversity between M segments of phleboviruses.

The M segment, which encodes the envelope proteins, was notably more saturated. We hypothesize that this increased variability can favor the escape of the virus from the hosts’ immunity and allow its occupation of different ecological niches. This appears to be confirmed by the different serological reaction in humans and animals of the three PONTV bearing different M segments [32,40].

## 5. Conclusions

In this study, PONTV III was isolated from sandflies collected in the field. The genomic structure, nucleotide and amino acid sequence homology, and molecular genetics evolution analyses of the virus confirmed that it is a virus of the *Adana phlebovirus* species. This is a new isolation of PONTV III from sandflies. Further studies should examine the host and epidemiological range of the virus in the field, especially whether the virus can infect and be pathogenic for humans, livestock, and other animals.

## Figures and Tables

**Figure 1 viruses-15-00422-f001:**
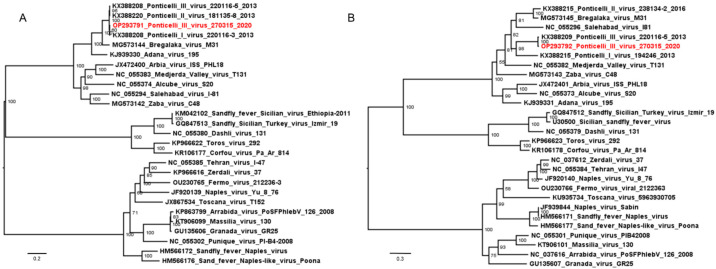
Mid-point rooted ML phylogenetic tree obtained by sequences of L segments (**A**), and M segments (**B**), of the viruses isolated from sandflies in the Mediterranean and Middle East. Virus isolated in this study is shown as red. Bootstrap values are shown near the nodes.

**Figure 2 viruses-15-00422-f002:**
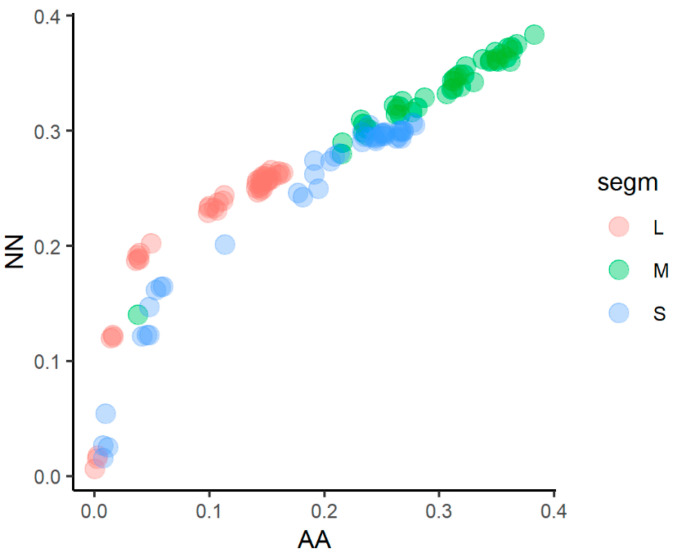
Nucleotide and amino acid uncorrected p-distance between viruses of the Salehabad complex.

**Figure 3 viruses-15-00422-f003:**
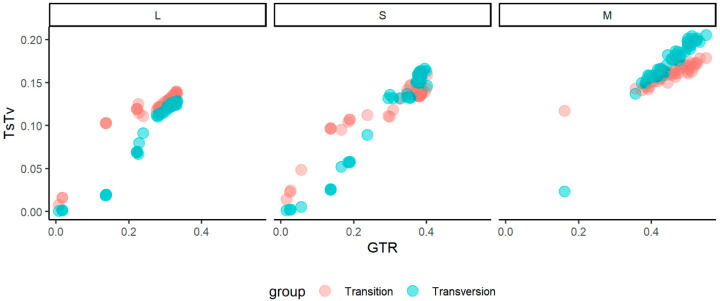
Saturation plots showing transitions and transversions versus GTR distance of the three genomic sequences of the Ponticelli, I Ponticelli II, and Ponticelli III viruses.

**Table 1 viruses-15-00422-t001:** Sandflies species composition in Casalgrande locality.

	18 June 2019	30 July 2019	13 August 2019	27 August 2019	10 September 2019	24 September 2019	4 September 2020	Total
*Phlebotomus perfiliewi*	2	40	15	5	10			72
*Phlebotomus perniciosus*	1							1
*Phlebotomus* spp.	0	154	53		67	0	135 *	409
Total	3	194	68	5	77	0	135	482

* Isolation on cell culture, 6 pools tested, 1 positive.

**Table 2 viruses-15-00422-t002:** Nucleotidic and amino acidic p distance between L (lower right) and M segments (upper left). According to the ICTV rule, virus with an aminoacid diversity of L segment < 0.05 can be considered the same species.

Virus	Abb.	ADAV	PONTVI	PONTVII	PONTVIII	BREV	SALV	ZABAV	ARBV	MVV
Adana virus 195	ADAV		0.34–0.33	0.34–0.31	0.35–0.32	0.35–0.31	0.36–0.34	0.34–0.32	0.31–0.27	0.35–0.32
Ponticelli I virus 220116–3/2013	PONTV I	0.19–0.04		0.3–0.23	0.29–0.22	0.3–0.24	0.32–0.26	0.35–0.32	0.37–0.36	0.33–0.27
Ponticelli II virus 181135–8/2013	PONTV II	0.19–0.04	0.02–0		0.3–0.24	0.14–0.04	0.31–0.23	0.34–0.31	0.36–0.34	0.32–0.26
Ponticelli III virus 270315/2020	PONTV III	0.19–0.04	0.02–0	0.01–0		0.3–0.24	0.31–0.26	0.34–0.31	0.37–0.36	0.32–0.28
Bregalaka virus M31	BREV	0.19–0.04	0.12–0.02	0.12–0.01	0.12–0.02		0.31–0.23	0.34–0.31	0.36–0.34	0.32–0.27
Salehabad virus I–81	SALV	0.25–0.14	0.26–0.14	0.25–0.14	0.26–0.14	0.25–0.15		0.36–0.32	0.38–0.38	0.33–0.29
Zaba virus C48	ZABAV	0.25–0.15	0.26–0.15	0.26–0.15	0.26–0.15	0.26–0.15	0.2–0.05		0.36–0.35	0.33–0.31
Arbia virus ISS PHL18	ARBV	0.26–0.15	0.26–0.15	0.26–0.15	0.26–0.15	0.26–0.15	0.23–0.1	0.23–0.11		0.36–0.35
Medjerda Valley virus T131	MVV	0.26–0.14	0.25–0.15	0.25–0.14	0.25–0.14	0.25–0.14	0.23–0.1	0.23–0.1	0.19–0.04	
Alcube virus S20	ACBV	0.27–0.15	0.26–0.16	0.26–0.16	0.26–0.16	0.26–0.16	0.24–0.11	0.24–0.11	0.24–0.11	0.23–0.1

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
