# Peer review of "New Isolation of Ponticelli III Virus (Bunyavirales: Phenuiviridae) in Emilia-Romagna Region, Italy"

_viruses, 2023, doi:10.3390/v15020422_

Round 1

Reviewer 1 Report

In this study, the authors are investigating the presence of phleboviruses virus in Sandflies collected in the Emilia-Romagna region in Italy. The authors collected a total 482 sandflies using CDC light traps baited with CO2 in 2019 and 2020. One pool of unknown phlebotomus species was found positive for CPE and Ponticelli III virus was identified through phylogenetic analysis. Authors isolated the virus and conducted phylogenetic analysis of the L, and M segment.

While the results from this article are relevant to the field and provide an insight on the circulation of Phleboviruses in the region, there are several weaknesses that need to be addressed

1)    Text and table 1 content show discrepancy in the number of sandflies tested

2)    While testing an unknown pool of sandfly was useful for the first detection of the virus, as performed in the 2013, this study could have benefited from first identifying sandflies to the species level and then testing pools of known species. This would allow to identify the potential vector of Ponticelli III virus.

3)    While the methods were published in a previous article it was hard to understand some of the method sections without having to go back to the original paper.

4)    More importantly, the phylogenetic analysis should have included the Ponticelli III virus sequences from the Ponticelli III 2013 isolate. While the Ponticelli I and II viruses were included, Ponticelli III was omitted.

Author Response

Response Letter

Manuscript number: 

Dear Editor,

We appreciate the opportunity to resubmit a revised version of our manuscript number viruses-1998451; entitled: New isolation of Ponticelli III Virus (Bunyavirales: Phenuiviridae) in Emilia-Romagna region, Italy.

We thank the reviewers for their careful reading and thoughtful comments. We have carefully taken their comments, corrections, and suggestions into consideration in preparing our revision, which has resulted in a paper that is clearer and more compelling. All modifications are highlighted in the attached manuscript.

Sincerely yours,

Response: All requested revisions have been made. Please see our responses below for every single comment and question.  

Responses to reviewers

Reviewer #1

Comment: in this study, the authors are investigating the presence of phleboviruses virus in Sandflies collected in the Emilia-Romagna region in Italy. The authors collected a total 482 sandflies using CDC light traps baited with CO2 in 2019 and 2020. One pool of unknown phlebotomus species was found positive for CPE and Ponticelli III virus was identified through phylogenetic analysis. Authors isolated the virus and conducted phylogenetic analysis of the L, and M segment.

While the results from this article are relevant to the field and provide an insight on the circulation of Phleboviruses in the region, there are several weaknesses that need to be addressed

Response: Thank you for reviewing our work. We appreciate your comments, corrections, and suggestions. You will find below the response for every comment made.
Comment 1: I Text and table 1 content show discrepancy in the number of sandflies tested
Response: Thank you for your comment. We corrected the number of specimens tested. Please see lines 146-151.

Comment 2: While testing an unknown pool of sandfly was useful for the first detection of the virus, as performed in the 2013, this study could have benefited from first identifying sandflies to the species level and then testing pools of known species. This would allow to identify the potential vector of Ponticelli III virus.

Response:  Thank you for your pointed comment and suggestion. Indeed, testing and identifying sand flies to the species level would allow us to identify the potential vector of Ponticelli III virus. In order to identify sand flies, however, a clarification step is needed which would make it impossible to test identified sandflies. Another possible method is dissection, but we prefer not to apply this method to avoid cross contamination of the samples or loss of viral positivity.

Comment 3: While the methods were published in a previous article it was hard to understand some of the method sections without having to go back to the original paper.

Response: Thank you for your comment. These changes have been made with more details and clarification of method section in the revised version of the manuscript.

Comment 4: More importantly, the phylogenetic analysis should have included the Ponticelli III virus sequences from the Ponticelli III 2013 isolate. While the Ponticelli I and II viruses were included, Ponticelli III was omitted.

Response: Thank you for your comment. We included the Ponticelli III virus sequences from the Ponticelli III 2013 isolate in the phylogenetic analysis. Please see the attached version of the manuscript.

Reviewer 2 Report

COMMENTS TO THE AUTHOR: 

The manuscript "New isolation of PonticelliIIIVirus (Bunyavirales: Phenuiviridae) in Emilia-Romagna region, Italy" addresses an interesting and under-researched topic, therefore it represents an interesting and valuable target. Nevertheless, the manuscript needs revision. A grammar review and a review technical English by an expert in this field of virology and a native speaker. 

  1. This sentence is not clear, check in the Instructions for Authors, how the abstract should be written according to the journal. 

  1. Check the term used, so the text refers to a type of methodology not described in item 2 (materials and methods), line 24. 

  1. Check the term used, that word is not correct, so the text refers to a type of methodology not described in item 2 (materials and methods). Line 24. 

  1. This sentence is not clear, check in the Instructions for Authors, how the abstract should be written according to the journal. Line 25-26. 

  1. Check the term used, this word does not make sense in the sentence and it is leaving the sentence meaningless, it must be rewritten, line 28-29. 

  1. It seems that this contains a series of three or more words, phrases, or clauses. Consider inserting a comma to separate the elements. “hard-to-read textline 48-51. 

  1. Check the spelling error, line 70. 

  1. Check grammar errors, and passive voice misuse, line 71. 

  1. I see the need to rewrite this sentence: because the authors report the isolation of a new Ponticelli III virus, not a new isolation method, lines 74-76. 

  1. The authors should rewrite this passage because it is not clear, it does not perfectly describe the idea of the sentence, making it difficult to understand the procedure performed. Lines 81-82. 

  1. The authors need to rewrite this sentence because the language used is not technical, it is not clear and difficult to understand. They need to describe this method more clearly and in more detail. Line 83-87. 

  1.  This sentence is duplicated or misspelled. The authors should review this sentence to verify. Line 96-98. 

  1. The authors should revise and rewrite this sentence because the term "cloned" does not look good in this sentence. It is not the correct term that should be used. Lines 99-100. 

  1. The authors describe the analysis method for the S segment, but in figure 1 they present the results with a tree of the alignment of the L and M segments, which is not written in Materials and Methods, and they do not present the result for the S segment described here in Materials and Methods. Lines 124-125. 

  1. The authors should describe the two genes of the S segment frame of reading to sense and o antisenses. Line 124. 

  1. The reference position is not suitable. Lines 130, and 133. 

  1. The authors need to rewrite this entire paragraph, it is unclear and makes it difficult to understand the result presented. Since 409 divided by 25-specimen gives about 16 pools. Only one pool was positive? The result presented here is not described according to the materials and methods, such as 7 days of observation of CPE and purification of isolates by the plaque method. I suggest that the authors present the results more descriptively. Lines 142-152.

Author Response

Response Letter

Manuscript number:  viruses-1998451

Dear Editor,

We appreciate the opportunity to resubmit a revised version of our manuscript number viruses-1998451; entitled: New isolation of Ponticelli III Virus (Bunyavirales: Phenuiviridae) in Emilia-Romagna region, Italy.

We thank the reviewers for their careful reading and thoughtful comments. We have carefully taken their comments, corrections, and suggestions into consideration in preparing our revision, which has resulted in a paper that is clearer and more compelling. All modifications are highlighted in the attached manuscript.

Sincerely yours,

Response: All requested revisions have been made. Please see our responses below for every single comment and question.  

Responses to reviewers

Reviewer #2

Comment to the authors: COMMENTS TO THE AUTHOR: 

The manuscript "New isolation of PonticelliIIIVirus (Bunyavirales: Phenuiviridae) in Emilia-Romagna region, Italy" addresses an interesting and under-researched topic, therefore it represents an interesting and valuable target. Nevertheless, the manuscript needs revision. A grammar review and a review technical English by an expert in this field of virology and a native speaker. 

Response: Thank you for reviewing our work. We are greatly appreciate your comments, corrections, and suggestions. We conducted an English review of our paper by a native speaker and an expert in the field of virology to improve its quality. All of the requested revisions have been addressed. Additionally, all of the provided suggestions were considered without exception. You will find below the response for every comment made.

Comment 1: This sentence is not clear, check in the Instructions for Authors, how the abstract should be written according to the journal

Response:  Thank you for your comment. Could you please specify which sentence and which line? Thanks in advance.

Comment 2 Check the term used, so the text refers to a type of methodology not described in item 2 (materials and methods), line 24

Response: Thank you. This change has been made. Please see the modification in line 25 of the revised version of the manuscript.

Comment 3: Check the term used, that word is not correct, so the text refers to a type of methodology not described in item 2 (materials and methods). Line 24. 

Response: Thank you. We made modifications in this section according to your comments. Please see the revised version of the manuscript.

Comment 4: This sentence is not clear, check in the Instructions for Authors, how the abstract should be written according to the journal. Line 25-26. 

Response: Thank you. We modified this section according to the Instructions for Authors. Please see the revised version of the manuscript.

Comment 5: Check the term used, this word does not make sense in the sentence, and it is leaving the sentence meaningless, it must be rewritten, line 28-29.

Response: Thank you. This change has been made. Please see lines 28 to 30.

Comment 6: It seems that this contains a series of three or more words, phrases, or clauses. Consider inserting a comma to separate the elements. “hard-to-read text” line 48-51. 
Response: Thank you. This section has been rewritten. Please see the revised version of the manuscript.

Comment 7: Check the spelling error, line 70

Response:  Thank you. This correction has been made. Please see the revised version of the manuscript.

Comment 8: Check grammar errors, and passive voice misuse, line 71. 

Response: Thank you for your comments. These corrections have been made. Please see the revised version of the manuscript.

Comment 9: I see the need to rewrite this sentence: because the authors report the isolation of a new Ponticelli III virus, not a new isolation method, lines 74-76. 

Response: Thank you for your comments. This sentence has been rewritten. Please see lines 74-76 in the revised version of the manuscript.

Comment 10: The authors should rewrite this passage because it is not clear, it does not perfectly describe the idea of the sentence, making it difficult to understand the procedure performed. Lines 81-82.

Response: Thank you for your comments. We rewrote the passage to articulate the idea more clearly. Please see lines 81-82 in the revised version of the manuscript.

Comment 11: The authors need to rewrite this sentence because the language used is not technical, it is not clear and difficult to understand. They need to describe this method more clearly and in more detail. Line 83-87. 

Response: Thank you for your comments. We rewrote the entire paragraph to make it clearer and to more clearly describe the method. Please see lines 81-87 in the revised version of the manuscript.

Comment 12:  This sentence is duplicated or misspelled. The authors should review this sentence to verify. Line 96-98. 

Response: Thank you for your comments. We removed the duplicated sentence. Please the revised version of the manuscript.

Comment 13: The authors should revise and rewrite this sentence because the term "cloned" does not look good in this sentence. It is not the correct term that should be used. Lines 99-100. 

Response: Thank you for your comments. This change has been made. Please see lines 96 and 98 in the revised version of the manuscript.

Comment 14: The authors describe the analysis method for the S segment, but in figure 1 they present the results with a tree of the alignment of the L and M segments, which is not written in Materials and Methods, and they do not present the result for the S segment described here in Materials and Methods. Lines 124-125. 

Response: Thank you for your comments. The concatenated sequences of the two genes of the S segment were used for the saturation plot. We added the analysis of Nc and Ns genes of S segment in supplemental material and highlighted this in line 124 in the revised version of the manuscript.

Comment 15: The authors should describe the two genes of the S segment frame of reading to sense and o antisenses. Line 124. 

Response: Thank you for your comments.  In the introduction section we elaborated more that the two genes of the S segment are in antisense (line 51). Please see the revised version of the manuscript.

Comment 16: The reference position is not suitable. Lines 130, and 133. 

Response: Thank you for your comment. The reference positions have been updated. Please the revised version of the manuscript.

Comment 17: The authors need to rewrite this entire paragraph, it is unclear and makes it difficult to understand the result presented. Since 409 divided by 25-specimen gives about 16 pools. Only one pool was positive? The result presented here is not described according to the materials and methods, such as 7 days of observation of CPE and purification of isolates by the plaque method. I suggest that the authors present the results more descriptively. Lines 142-152.

Response: Thank you for your comments. We rewrote the entire paragraph to make it more clear. Please lines 139- 149 in the revised version of the manuscript

Reviewer 3 Report

Review of New isolation of Ponticelli III Virus (Bunyavirales: 2 Phenuiviridae) in Emilia-Romagna region, Italy, by Daoudi et al. submitted to Viruses to be published as a Communication

This "communication" described the isolation and genetic characterization of a new isolation of a strain of Ponticelli III virus. This manuscript has been submitted to a previous revision process and only minor points were identified, requiring minor revisions, as listed below.

Line 19: Correct " sandflies phleboviruses" to "sandfly-borne phleboviruses".

Line 20: Correct "are pathogenic in humans" are "are human pathogens".

Line 37: the Phenuiviridae family is composed of 20 genera, not 13 (https://ictv.global/taxonomy).

Line 37: add "the members of" before "three".

Line 52-53: Correct to either "The structure of all phleboviruses is similar..." OR "Phlebovituses are morphologically similar". The current sentence is a mix of the two.

Line 57: Replace "were not" by "have not yet been".

Line 104: Correct " Super Script H" to " SuperScript™ II".

Line 109 and 117: Correct "Gen Bank" to "GenBank". The "GB" abbreviation should have been mentioned in line 109 (following GenBank).

Line 142: The sentence " Only 2019 only some specimens were identified" is awkward and requires a correction.

Line 144: A space is missing before " The 135".

Line 154: Use GenBank or GB, but not both. The abbreviation has been defined before.

In table 2 "," associated with numerical values should be replaced by "."...(including the table caption).

Line 175: Correct "obtained by" to "obtained for".

Line 209: Correct " obtained by sequences" to "obtained as the result of the analysis of of L segments (A) and M segments (B) sequences". The same correction should be done in the caption of supplementary figure 1.

Line 187/190/193: The "NN" abbreviation has not been defined.

Caption to Figure 1: I presume the values indicated correspond to bootstrap percentages, but they could also correspond to aLRT values. For the sake of correction, the authors should indicate explicitly what do they correspond to.

Lines 231-232: A ")" is missing, probably after "bats".

Line 256: "Substitution" should be corrected to plural.

Author Response

Response Letter

Manuscript number: 

Dear Editor,

We appreciate the opportunity to resubmit a revised version of our manuscript number viruses-1998451; entitled: New isolation of Ponticelli III Virus (Bunyavirales: Phenuiviridae) in Emilia-Romagna region, Italy.

We thank the reviewers for their careful reading and thoughtful comments. We have carefully taken their comments, corrections, and suggestions into consideration in preparing our revision, which has resulted in a paper that is clearer and more compelling. All modifications are highlighted in the attached manuscript.

Sincerely yours,

Response: All requested revisions have been made. Please see our responses below for every single comment and question.  

Reviewer #3

This "communication" described the isolation and genetic characterization of a new isolation of a strain of Ponticelli III virus. This manuscript has been submitted to a previous revision process and only minor points were identified, requiring minor revisions, as listed below.

Response: Thank you for reviewing our work. We appreciate your comments, corrections, and suggestions. You will find below the response for every comment made.

Line 19: Correct " sandflies phleboviruses" to "sandfly-borne phleboviruses".

Response: Thank you. This correction has been made. Please see the revised version of the manuscript.

Line 20: Correct "are pathogenic in humans" are "are human pathogens".

Response: Thank you. This correction has been made. Please see the revised version of the manuscript.

Line 37: the Phenuiviridae family is composed of 20 genera, not 13 (https://ictv.global/taxonomy).

Response: Thank you. This correction has been made. Please see the revised version of the manuscript.

Line 37: add "the members of" before "three".

Response: Thank you. This correction has been made. Please see the revised version of the manuscript.

Line 52-53: Correct to either "The structure of all phleboviruses is similar..." OR "Phlebovituses are morphologically similar". The current sentence is a mix of the two.

Response: Thank you. This correction has been made. Please see the revised version of the manuscript.

Line 57: Replace "were not" by "have not yet been".

Response: Thank you. This correction has been made. Please see the revised version of the manuscript.

Line 104: Correct " Super Script H" to " SuperScript™ II".

Response: Thank you. This correction has been made. Please see the revised version of the manuscript.

Line 109 and 117: Correct "Gen Bank" to "GenBank". The "GB" abbreviation should have been mentioned in line 109 (following GenBank).

Response: Thank you. This correction has been made. Please see the revised version of the manuscript.

Line 142: The sentence " Only 2019 only some specimens were identified" is awkward and requires a correction.

Response: Thank you. This correction has been made. Please see the revised version of the manuscript.

Line 144: A space is missing before " The 135".

Response: Thank you. This correction has been made. Please see the revised version of the manuscript.

Line 154: Use GenBank or GB, but not both. The abbreviation has been defined before.

In table 2 "," associated with numerical values should be replaced by "."...(including the table caption).

Response: Thank you. This correction has been made. Please see the revised version of the manuscript.

Line 175: Correct "obtained by" to "obtained for".

Response: Thank you. This correction has been made. Please see the revised version of the manuscript.

Line 209: Correct " obtained by sequences" to "obtained as the result of the analysis of of L segments (A) and M segments (B) sequences". The same correction should be done in the caption of supplementary figure 1.

Response: Thank you. This correction has been made. Please see the revised version of the manuscript.

Line 187/190/193: The "NN" abbreviation has not been defined.

Response: Thank you. This correction has been made. Please see the revised version of the manuscript.

Caption to Figure 1: I presume the values indicated correspond to bootstrap percentages, but they could also correspond to aLRT values. For the sake of correction, the authors should indicate explicitly what do they correspond to.

Response: Thank you. This correction has been made. Please see the revised version of the manuscript.

Lines 231-232: A ")" is missing, probably after "bats".

Response: Thank you. This correction has been made. Please see the revised version of the manuscript.

Line 256: "Substitution" should be corrected to plural.

Response: Thank you. This correction has been made. Please see the revised version of the manuscript.

Round 2

Reviewer 2 Report

Dear Authors,

The manuscript "New isolation of Ponticelli III Virus (Bunyavirales: Phenuiviridae) in the region of Emilia-Romagna, Italy" addresses an interesting and little researched topic. Therefore, it represents an interesting and valuable target. In this way, the manuscript will be of great value to the scientific community.

1- I recommend that authors change the word "inoculated" to the word "isolada", line 25.

Kind regards,

Author Response

Response Letter

Manuscript number: 

Dear Editor,

We appreciate the opportunity to resubmit a revised version of our manuscript number viruses-1998451; entitled: New isolation of Ponticelli III Virus (Bunyavirales: Phenuiviridae) in Emilia-Romagna region, Italy.

We thank the reviewers for their careful reading and thoughtful comments. We have carefully taken their comments, corrections, and suggestions into consideration in preparing our revision, which has resulted in a paper that is clearer and more compelling. All modifications are highlighted in the attached manuscript.

Sincerely yours,

Response: All requested revisions have been made. Please see our responses below for every single comment and question.  

Responses to reviewers

Reviewer #2

The manuscript "New isolation of Ponticelli III Virus (Bunyavirales: Phenuiviridae) in the region of Emilia-Romagna, Italy" addresses an interesting and little researched topic. Therefore, it represents an interesting and valuable target. In this way, the manuscript will be of great value to the scientific community.

  • I recommend that authors change the word "inoculated" to the word "isolada", line 25

Response: Thank you for reviewing our work. We appreciate your comments, corrections, and suggestions. These changes have been made, please see the revised version of the manuscript.